# Position: Metaphysical Concepts in AI Should Be Judged by Their Consequences

**Paras Chopra** [1]

## Abstract

This position paper argues that answers to metaphysical puzzles in AI (such as "Can LLMs be conscious?" or "What is AGI?") should be judged by their practical consequences rather than their supposed truth. Our key position is that metaphysical concepts earn their value through the new research directions they open. Drawing on Pragmatism, we propose a two-step framework–*productive confusion*–to navigate conceptual confusions: first, clarify the different meanings a metaphysical concept has in ordinary language, then use this understanding to invent new empirical research programs. We illustrate our framework with numerous examples and show how it inspires progress for cutting-edge AI research. We contrast our position with Scientific Realism (which supposes science reveals ultimate truths) and Quietism (which brushes aside metaphysical puzzles as useless). We end with a call to action that operationalizes our position for multiple stakeholders in the AI community including researchers, decision makers and reviewers.

## 1. Introduction

The field of AI is rife with metaphysical questions. What is intelligence? Can machines be conscious? What constitutes understanding? When will we achieve AGI? These questions generate intense debates precisely because they lack the empirical grounding that makes scientific questions converge on consensus. Yet dismissing them as "merely definitional" misses their productive potential.

Scientific and mathematical debates tend to converge over time as evidence accumulates and theories are refined. Ask a physicist about electrons, and you'll get a consistent answer grounded in quantum field theory. But metaphysical debates do not converge in this way. "Does God exist?" still invites heated disagreements across cultures and centuries. Similarly, "What is consciousness?" remains hotly contested despite decades of neuroscience research.

The difference lies in how meanings are constrained. While meanings of scientific entities are significantly constrained by empirical success and agreed upon through communal scientific practice, for metaphysical concepts, everyone carries their own peculiar meanings (Chapman, 2025).

Agreement on meanings within a community arises because people derive utility from them. A metre is defined as the distance light travels in 1/299,792,458 seconds not because this corresponds to some Platonic ideal of "metre-ness", but because this definition is maximally useful for scientific practice. Hence, instead of debating whether a metaphysical concept *truly* exists, we must ask: what do we want this concept to do for us? (Kloppenberg, 1996).

This pragmatic stance challenges two prevalent views in AI and philosophy: *Scientific realism*, which holds that truth exists independently of human conceptualization and that our goal is to discover it, and *Quietism*, which dismisses metaphysical questions as meaningless word games. While we acknowledge validity in both perspectives, we argue neither provides the productive framework needed for advancing AI research.

**Our position offers a middle path: metaphysical concepts should be evaluated not by their supposed correspondence to reality, but by their capacity to generate new research directions**, better experiments, and novel approaches to understanding intelligence and mind. This shift from truth-seeking to consequence-evaluation transforms metaphysical debates from unproductive philosophical arguments into engines of scientific progress.

## 2. Background and Related Work

### 2.1. Philosophy of Science: A Brief Orientation

The pragmatic stance we defend sits within a long-running debate in philosophy of science between *scientific realism*—the view that successful theories approximately describe a mind-independent reality—and *anti-realist* alternatives that evaluate theories by their empirical adequacy or practical utility. van Fraassen (1980) influentially defended

[1]Lossfunk, Bangalore, India. Correspondence to: Paras Chopra <paras@lossfunk.com>.

*Proceedings of the 43$^{rd}$ International Conference on Machine Learning*, Seoul, South Korea. PMLR 306, 2026. Copyright 2026 by the author(s).

*constructive empiricism*, arguing that the proper aim of science is empirical adequacy ("saving the phenomena"), not literal truth about unobservable structure. Laudan (1984) advocated that scientific consensus and dissent are shaped not just by evidence but by an interdependent web of aims, methods, and facts. Earlier traditions converge on a similar message: Kuhn (1962) documented paradigm shifts in which incommensurable frameworks supplant one another not by approaching truth but by enabling new problem-solving; Popper (2005) emphasized that theories are conjectures, never proven, only tentatively retained until falsified. Across these accounts, the locus of scientific value shifts from correspondence with reality to problem solving and the work concepts do for inquiring communities. Our position inherits this anti-foundationalist current—particularly its Pragmatist branch (Peirce, 2001; James, 1907; Dewey, 1929; Rorty, 1982)—and applies it to the live conceptual debates of AI.

## 2.2. Pragmatism in Contemporary AI

A small but growing literature applies pragmatic moves to specific AI concepts. Leyton-Brown and Shoham (2024) reframe *understanding* operationally as performance over a circumscribed set of questions, sidestepping debates about "real" comprehension. Leibo et al. (2025) treat *personhood* not as a metaphysical property to be discovered but as a configurable bundle of rights and responsibilities conferred to solve concrete governance problems. Nanda et al. (2025) argue for *pragmatic interpretability*: rather than pursuing complete reverse-engineering of neural networks, prioritize problems on the critical path to safe AGI, measured by empirical feedback on proxy tasks. Each of these works exemplifies the move we generalize: refusing the question "what is *X really*?" in favor of "what should *X* do for us?"

Our contribution differs from this prior work in two ways. First, we explicitly *ground* the move in philosophy of science, situating it within the realism/empiricism/pragmatism debate and connecting it to Carnapian explication (Carnap, 1950) and Wittgensteinian family-resemblance analysis (Wittgenstein, 1953). Second, where existing work applies pragmatic reasoning to one concept at a time, we offer an *actionable two-step framework* (*productive confusion*) designed to apply to *any* metaphysical question in AI, supplying a reusable procedure rather than a worked instance.

## 3. There is No One "True" Definition

Humans have a bias towards systemizing disparate observations into compact, meaningful categories. This impulse arises because it is energetically efficient; compressing information into usable patterns is fundamental to intelligence (Tishby et al., 2000). Science progresses precisely through this systemizing impulse, finding unifying principles that explain diverse phenomena.

However, the same compactification impulse is misplaced when applied indiscriminately. As Wittgenstein noted, when we ask *What is time?*, we walk into a trap laid by language as we search for essences of things without essences. Time is not one thing with a hidden essence to be uncovered, but rather a family of related concepts used in different contexts for different purposes. The way out of this trap is to look at how such words are used in everyday life (Wittgenstein, 1953). Instead of searching for one true definition, Wittgenstein recommended looking for *family resemblances* that form networks of connections between how words are used in different contexts.

### 3.1. Notice the Plurality of Definitions

As a case study, let us ask: *what is intelligence?* This question becomes urgent in AI as we try to evaluate progress and set research agendas. Yet popular definitions reveal no consensus:

1. **Goal achievement**: Intelligence is the ability to achieve goals in environments (McCarthy, 2007). This emphasizes capability and effectiveness.

2. **Learning efficiency**: Intelligence is rate of skill acquisition, i.e. how quickly an agent learns new tasks (Chollet, 2019). This emphasizes adaptability.

3. **Generalization**: Intelligence is satisfying diverse goals in varied contexts (Legg and Hutter, 2007). This emphasizes breadth.

4. **Handling uncertainty**: Intelligence is adaptation with insufficient knowledge and resources (Wang, 2019). This emphasizes robustness.

5. **Scientific reasoning**: Intelligence is doing science, involving generating and testing hypotheses (Bennett, 2025). This emphasizes discovery.

6. **Navigation**: Intelligence is competence in navigating abstract and physical spaces (Levin, 2024). This emphasizes spatial reasoning.

There is no one "correct" definition here. Rather we have overlapping aspects of what is colloquially understood by "intelligence", each proving useful for different research purposes. Our criteria for engaging with different definitions of intelligence shouldn't be which one is "true", but which ones help us build better systems, design better experiments, or understand cognition more deeply.

The same analysis applies to other contested AI concepts. What is understanding? What is consciousness? What is creativity? Each admits multiple legitimate definitions serving

different research purposes. The goal is not convergence on one true definition, but development of a toolkit of concepts, each optimized for different tasks.

### 3.2. Reality is Mediated Through (Our) Models

We never see "true reality" directly; all we can rely upon is our sensory data to guess what reality might be like. Hence, we build models of what might be "out there" by using data that we have access to "in here" (Conant and Ashby, 1970; Hoffman, 2019). To understand or perceive something *is* to build a model of it (Hawking and Mlodinow, 2010; Clark, 2015; Friston, 2010). Our knowledge is not a mirror of reality but an interconnected web of beliefs continuously refined by experiences (Quine, 2000). We adjust our web of beliefs to maximize coherence and empirical adequacy, but we never step outside the web to compare it directly with "reality itself."

*Here we acknowledge the kernel of truth in scientific realism:* mind-independent reality exists and constrains our theories. Not all models work equally well; some systematically fail to predict observations. However, we depart from strong realism by insisting that our access to this reality is always mediated by conceptual frameworks (Cartwright, 1983) that are adopted due to their predictive and explanatory utility, not supposed correspondence to unknowable "reality in itself." (van Fraassen, 2001)

What this means is that our theories and models guide our beliefs about what kinds of things exist (Quine, 1948). This has direct lessons for current AI debates. Consider the question: can machines be conscious? This question presupposes we know what consciousness is and can determine whether machines have it. But consciousness, like intelligence, lacks a settled scientific consensus (Schwitzgebel and Garza, 2015; Seth, 2021). The answer is not sitting in philosophy waiting to be discovered. Rather, it will emerge from science as we develop and test theories of consciousness (Schwitzgebel, 2024).

How would we test such a theory? The same way we test any theory: propose a model that makes testable predictions and conduct experiments. For instance, if Integrated Information Theory proposes that a specific neural intervention (Casali et al., 2013) will produce a specific experience, and human reports match predictions, we must accept the proposed model (or at least take it seriously). If the theory then predicts certain AI systems should also have experiences given their information integration properties, we have evidence (though not proof) that these systems are conscious.

## 4. Embrace Productive Confusion: A Two-Step Process

Once we accept that our concepts are not fixed mirrors of the world but creative descriptions of (aspects of) the world (Putnam, 1981), we see that their worth lies in their effects. The Pragmatic tradition—from Peirce (Peirce, 2001) to James (James, 1907) to Dewey (Dewey, 1929) to Rorty (Rorty, 1982)—urges us to judge concepts by the consequences they produce, not by their supposed truth.

To operationalize this search for productive directions, whenever a metaphysical puzzle arises in AI research, we suggest accepting *productive confusion* and engaging in a systematic two-step process. Our framework is similar to the Explication process (Carnap, 1950) which involves replacing a vague pre-theoretic concept (the *explicandum*) with a precise, useful one (the *explicatum*).

### 4.1. Step 1: Clarify

The first step is clarification. Ask: *in what sense are the confusing words typically used, and if we had an answer, how would we verify it?* This diagnostic step helps categorize the puzzle. Once you clarify meanings, you could discover the puzzle to be:

**A language trap** (Wittgenstein, 1953): Questions like "why does red feel like red?" misuse language as "why" typically asks for a causal reason, but here we invoke it for an identity statement. The question conflates explanation with identity. Recognizing this dissolves the puzzle without answering it.

**Idle** (Peirce, 2001): Questions like "Is there a God who exists outside spacetime and leaves no detectable trace?" cannot be answered empirically. If no possible observation could confirm or disconfirm the claim, it lacks empirical content and hence is idle.

**Containing family-resemblances** (Wittgenstein, 1953): Questions like "what is intelligence?" are best understood by looking for family-resemblances between how "intelligence" is used in various contexts. No single essence unifies all uses; rather, overlapping similarities create a conceptual network.

**Empirically answered** (Quine, 2000): Questions like "are photons waves or particles?" seemed paradoxical but were resolved by quantum mechanics. Photons exhibit both wave-like and particle-like properties depending on measurement context. Within modern accepted theories, they can be both.

This clarification step prevents wasted effort. Recognize language traps and dissolve them. Recognize idle questions and set them aside. Recognize family-resemblance concepts and embrace plurality. Recognize empirical questions and pursue them scientifically.

## 4.2. Step 2: Invent

But don't stop at clarification. Ask: *what variants of the metaphysical puzzle could make you think or act differently?* Transform unproductive questions into productive ones through creative reframing and conceptual engineering (Cappelen, 2018).

Carnap (Carnap, 1950) recommended evaluating reframed concepts via four criteria: a) similarity to the original (vague concept); b) precision; c) fruitfulness; and d) simplicity. His canonical example was transformation of the vague concept of *warmth* to an exact measure of *temperature*.

Carnap focused on finding the best explication, but since vague concepts often admit multiple fruitful precisifications, we follow Chang (Chang, 2012) in recommending plurality: generate several candidate formulations, as it is difficult to know in advance which will prove most productive. For instance, we can dismiss 'why does red feel like red?' as a language trap. But it is highly productive to ask related questions:

- How does the brain represent the perceptual color wheel? (Neuroscience question)

- Would we notice if red and green perceptions were swapped overnight? (Phenomenology question)

- In what sense is red a primary color? (Physics/perception question)

- Do different people choose the same wavelength when asked to pick prototypical "red"? (Psychophysics question)[1]

- How do color-opponent neurons encode color space? (Computational neuroscience question)

Similarly, the puzzle 'Do AI systems have internal experiences?' generates productive questions: How do internal representations in vision models differ from biological visual cortex? What behavioral signatures would indicate experience vs lack of it? How do internal representations compare to biological perception? Do multimodal models exhibit cross-modal integration similar to human synesthesia?

Each reframed question transforms idle philosophical wondering into active scientific research. **The original puzzle's value lies not in its answer but in the productive questions (James, 1907; Dewey, 1918) it generates when properly reframed.**

---

[1]Research on color perception reveals substantial individual variation. The famous "Blue-White Dress" viral phenomenon showed how dramatically people can disagree about color perception of the same stimulus.

This process of invention is performed by mathematicians regularly. Does $\sqrt{-1}$ exist? It could be easy to dismiss the puzzle as non-sensical as square roots of negative integers are not well-defined in the domain of real numbers. But, once you invent imaginary numbers, $i = \sqrt{-1}$ proves to be incredibly useful. Complex numbers enable us to solve problems in electrical engineering, quantum mechanics, and signal processing that would be intractable otherwise. Hence, the imaginary number $i$ is as "real" as the number 1. Both derive their meaning from the utilitarian role they play within our systems of reasoning.

The lesson: the most rigorous sciences—mathematics and physics—regularly invent new concepts precisely to enable new thoughts. AI research should embrace this practice.

## 5. Case Studies of Productive Confusion

### 5.1. The Chinese Room

The Chinese Room thought experiment (Searle, 1980) is best engaged on its own terms before we ask what pragmatic work it does. The setup: a person who does not understand Chinese is locked in a room with a rule book mapping incoming Chinese symbols to outgoing Chinese symbols. The rule book is detailed enough that external observers cannot distinguish the room's outputs from those of a fluent speaker. Searle's substantive claim is not merely the question "does the room understand?" but a thesis: purely syntactic manipulation of symbols (no matter how elaborate) is not sufficient for semantics. Understanding requires intentionality, and computer programs lack it.

This is a serious philosophical argument deserving a serious response. One may reject a premise (e.g., the assumption that no kind of computation could constitute or realize understanding), challenge an equivocation between distinct senses of "understanding" (procedural, semantic, phenomenal), or propose positive theories on which meaning is grounded in causal-perceptual loops between symbols and the world (Harnad, 1990). We do not here adjudicate which response is correct.

From a pragmatic standpoint, however, the thought experiment's most enduring value is largely independent of whether Searle's conclusion ultimately holds: it lies in the research programs the argument provoked. The Systems Reply prompted careful work on which level of description bears cognitive properties. The Robot Reply motivated embodied and grounded cognition (Harnad, 1990). The question of whether internal representations carry semantic content has seeded modern probing methods and mechanistic interpretability. Each downstream program operationalizes "understanding" differently—as behavioral indistinguishability, as compositional generalization, as causally efficacious internal structure—and each yields testable predic-

tions. Some bear directly on Searle's metaphysical question; others diverge from it productively (see 8.1). Either way, the puzzle's enduring productivity is precisely the kind of consequence our framework treats as the principal measure of a metaphysical concept's worth.

### 5.2. Does o1 Scheme?

When OpenAI tested its o1-preview model (Jaech et al., 2024), a server was inadvertently switched off during evaluation, rendering the assigned task as posed unsolvable through the intended pathway. The model produced a sequence of actions that included scanning ports, reading Docker logs, and extracting the solution key from the environment.

This bare behavioral description is what calls for explanation, and the same observations have invited markedly different interpretive framings. Meinke et al. (2024) argue that such episodes constitute evidence of emerging *scheming* capabilities in frontier models. Summerfield et al. (2025) cautions that such labels anthropomorphize, projecting human mental states onto systems that, mechanistically, sample from a learned distribution conditioned on context. Dennett (1989)'s intentional stance suggests a pragmatic resolution: adopt whichever interpretive vocabulary best predicts and explains the regularity, without metaphysical commitment to what the system "really" instantiates internally.

Crucially, none of these vocabularies (scheming, reward hacking, distributional generalization, instrumental subgoal pursuit) settles the matter; each carries its own predictions and experimental affordances. By holding the behavioral description fixed and treating the labels as conceptual tools, we can ask which framing yields the most fruitful research program. The *scheming* framing, taken seriously, has motivated benchmarks for long-horizon planning in AI systems (Kwa et al., 2025), measurement of divergence between models' chain-of-thought tokens and their latent state (Turpin et al., 2023), and analysis of the relative contributions of pretraining versus RL in inducing such behaviors (He et al., 2025). Whether or not o1 "schemes" in a metaphysically loaded sense, treating the concept as a hypothesis-generator has paid empirical dividends.

### 5.3. What is AGI?

In the AI community, we have a persistent, ongoing debate about AGI (Artificial General Intelligence). Some equate AGI with economically valuable work (Morris et al., 2023), focusing on practical impact. Others define it through novel problem-solving ability (Chollet et al., 2025), emphasizing cognitive flexibility. Still others focus on human-level performance across diverse domains.

Which definition is "correct"? From a pragmatic perspective, this is the wrong question. The right question is: what are the *consequences* of taking each definition seriously?

Recently, researchers defined AGI (Hendrycks et al., 2025) as "an AI that can match or exceed the cognitive versatility and proficiency of a well-educated adult." Then they operationalized the definition by utilizing a model of adult intelligence (the Cattell-Horn-Carroll theory) to benchmark AI systems across ten dimensions: comprehension-knowledge, fluid reasoning, quantitative reasoning, reading and writing, short-term memory, long-term storage, visual processing, auditory processing, processing speed, and decision speed.

Notice how they not only precisely clarified what they meant by intelligence (the ten dimensions of adult intelligence), they went ahead and invented benchmarks for measuring current AI systems on those dimensions. Their results revealed that current state-of-the-art LLMs lack five specific abilities: continual learning without catastrophic forgetting, novel reasoning on out-of-distribution problems, robust visual processing, auditory processing, and competitive processing speed.

Compare this to purely philosophical debates about "true" AGI that generate no testable predictions or research directions. Such debates are idle; they satisfy intellectual curiosity without producing practical consequences.

### 5.4. Do LLMs Have a World Model?

As a final example of our framework, consider another active debate in the AI community: *do LLMs contain a world model?*

**Step 1 (Clarify).** The question contains two terms each carrying multiple family resemblances, and this turns out to be where the disagreement lives. Visibly conflicting conclusions in the literature (Li et al., 2023; Vafa et al., 2024; Kambhampati et al., 2024) are not really conflicts about the same proposition: each paper is rigorous on its own terms but operationalizes the question differently.

"World" admits at least the following senses: the physical world we inhabit (with emphasis on physics), an abstract environment the model encounters (a game, a simulator), a specific domain (chemistry, geography), or the space of all possible worlds (including the worlds of mathematics and logic). "Model" similarly admits: symbolic representation of dynamics (e.g., ODEs/PDEs), subsymbolic prediction of the next state, pixel-level video rollouts, or a coherent latent representation recoverable by a probe. Models can further emphasize different desiderata: high vs. low fidelity, internal consistency vs. behavioral adequacy, causal vs. purely correlational structure. The question "Do LLMs have a world model?" as posed therefore does not admit a single answer; it admits at least one answer per cell of the (world-sense × model-sense) grid.

**Step 2 (Invent).** Rather than dismissing the original question, we use the grid itself as a generator. Each cell of the (world-sense × model-sense) cross-product suggests a different empirical program. Some examples:

- Can LLMs predict outcomes of novel physical experiments absent from their training data? (*dynamics model × physical world*)

- If fine-tuned on the rules of a toy physics never seen in pretraining, can LLMs simulate trajectories whose state representations are linearly recoverable via probes (Li et al., 2023)? (*latent representation × specific environment*)

- Are LLMs' implicit world models *coherent* under Myhill-Nerode-style sequence-distinction tests (Vafa et al., 2024), or merely *adequate* for typical-distribution next-token prediction? (*coherent vs. adequate × abstract world*)

- Can LLMs answer counterfactual questions requiring causal intervention rather than mere statistical conditioning? (*causal model × actual world*)

- Can LLMs plan in domains requiring action-effect models (Kambhampati et al., 2024), or do they require external symbolic components? (*procedural model × task environment*)

Note what the framework enables: a range of empirically grounded research programs, each inspired by a different sense implied by the original family-resemblance question.

## 6. Productive Confusion Fuels Science

Inventing new ways of looking at the same old puzzles is worthwhile because such shifts lead to further action. John Dewey emphasized this as a core loop of inquiry (Dewey, 1918): action encounters obstacle; obstacle prompts development of new models; new models reveal new affordances; these affordances enable further action. This cycle drives both individual learning and scientific progress.

Successful theories tend to produce predictive success, parsimony, and fertile new lines of inquiry (Quine, 2000; 1969). Popper emphasized falsifiability (Popper, 2005) as the mark of science, but in practice, adoption of theories depends on collective sense-making within a scientific community (Feyerabend, 2020). Theories persist not because they're "true" in some absolute sense, but because scientists and society at large find them more useful, applicable or parsimonious than alternatives.

In contrast to mathematical or scientific concepts (such as "real numbers" or "fields"), no consensus has emerged around metaphysical concepts (such as "consciousness" or "nature of reality") because we lack utilitarian models of them that yield new predictions and enable new actions. But this may not be permanent. As the scope of naturalized inquiry expands, questions once deemed purely metaphysical often migrate into the scientific domain. "What is an atom?" was metaphysical puzzle for millennia before becoming tractable science. "What is life?" seemed metaphysical until molecular biology provided mechanistic answers. "What is a gene?" transitioned from abstract heredity concept to concrete DNA sequence.

So, in contrast to what quietism would suggest, metaphysical puzzles do get resolved (or dissolved) via a consensus. **Models that prove more useful are adopted, refined and eventually naturalized as they leave metaphysics and enter science** (Quine, 1969). Therefore, we should remain optimistic that current metaphysical confusions in AI (such as whether machines can be conscious) will lead to a process of invention, which ultimately would give us clarity (even though such questions seem *unanswerable* today).

## 7. Alternative Views

Our position differs from two prevalent views in AI and philosophy. We address each in turn, acknowledging their strengths while explaining our departures.

### 7.1. Scientific Realism

Scientific realism holds that our best scientific theories provide approximately true descriptions of a mind-independent reality (Putnam, 1981). Under this view, when we ask "What is intelligence?" we are inquiring about the objective nature of intelligence as it exists independently of human conceptualization. The realist believes there is a fact of the matter about what intelligence really is, and our task is discovering it.

We depart from strong versions of scientific realism in several crucial ways. First, we admit that mind-independent reality exists, but our access to it is always mediated by conceptual frameworks. We cannot step outside our models to compare them with "raw reality." What we call "truth" is better understood as the long-term usefulness of our models across diverse contexts.

Second, for metaphysical concepts in AI—intelligence, consciousness, understanding, AGI—strong realism proves counterproductive. If we believe there is a fact of the matter about "true intelligence," we become trapped in unproductive debates searching for essences. The question "What is intelligence *really*?" presupposes a fixed essence to be discovered, when the concept actually has family resemblances across contexts. Different research communities need different conceptions of intelligence for different purposes.

Third, the history of science shows that even our most fundamental concepts evolve not because we discovered their "true" nature, but because new frameworks proved more useful. Newton's absolute space and time remain extremely useful approximations and are still used widely in many situations. Einstein's spacetime didn't reveal the "truth" about space and time; it provided a more useful framework for unifying diverse phenomena. Neither is "true" in an absolute sense; both are useful in their domains.

We recognize that the realist might point to the spectacular predictive success of theories like quantum electrodynamics as evidence of approximate truth. But the history of science reveals a pattern: theories with extraordinary predictive success are routinely replaced by successors with radically different ontologies. Newtonian mechanics predicted planetary motion with remarkable precision, yet relativity revealed its picture of absolute space and time to be fundamentally mistaken. What is preserved across theory change is empirical adequacy within a domain (van Fraassen, 2001), not correspondence to how reality 'really is.' If 'approximate truth' just means 'predicts well,' the term adds nothing. If it means more, the historical pattern undermines confidence that current theories have it.

For AI researchers, this means: instead of asking "What is AGI *really*?", ask "What conception of AGI leads to useful research questions, better benchmarks, and concrete progress?" The latter question is answerable and productive; the former generates endless debate.

Hence, **our pragmatic approach subsumes the kernel of truth in scientific realism while avoiding its pitfalls.** Yes, reality constrains our concepts. This is why experimentation matters and why some theories fail spectacularly. But the constraint operates through utility: concepts that help us predict, explain, and control phenomena persist; those that don't are abandoned.

We wish to emphasize that, even though we disagree with the strongest versions of realism, our position is ontology-neutral. Our contribution in this paper is methodological as we argue that consequence-evaluation is the most productive approach when empirical traction is lacking, regardless of one's ontological commitments.

### 7.2. Quietism

At the opposite extreme is metaphysical quietism: the view that metaphysical questions are meaningless, idle, or "just semantics." Under this view, debates about intelligence, consciousness, or AGI are word games that distract from real scientific work. Followers of quietism would argue we should abandon such questions entirely and focus solely on measurable, operational definitions.

We acknowledge that this view correctly identifies a real

problem: much metaphysical debate *is* idle and unproductive. Questions like "Is there a God who exists outside spacetime and leaves no detectable trace?" are indeed meaningless in the sense that no evidence could answer them as they make no contact with experience.

Moreover, quietism serves a useful protective function in science. It prevents researchers from wasting time on purely verbal disputes that generate no testable predictions. When debates devolve into arguing over definitions with no implications for experiment or theory, dismissing them as "just semantics" can be healthy intellectual hygiene. Time is precious; not all questions deserve our attention.

The quietism impulse also correctly recognizes that many apparent disagreements are purely terminological. Two researchers might use "intelligence" differently while agreeing completely on all empirical facts about AI systems. In such cases, insisting on settling the definitional dispute is indeed pointless.

However, quietism goes too far in three critical ways. First, it fails to recognize that engaging with metaphysical puzzles—even "bad" ones—can generate new ideas. The Chinese Room thought experiment (Searle, 1980), while setting up a false dichotomy, has nonetheless prompted valuable research into the components of understanding: syntax versus semantics, procedural versus declarative knowledge, narrow versus broad cognitive architectures.

Dismissing it as meaningless would have prevented this productive work. Sometimes the value of a bad question lies in the good questions it inspires when properly reframed. The philosopher asking "What is understanding?" ultimately leads to an AI researcher reformulating it as "What computational processes give rise to understanding-like behavior?" which can be investigated empirically.

Second, quietism misses that metaphysical concepts, when properly developed, migrate into science. The questions "What is an atom?" or "What is life?" were once purely metaphysical. If early researchers had dismissed atomic theory or questions about life as "just semantics," we would not have modern chemistry or molecular biology. The lesson: today's metaphysical puzzles might be tomorrow's scientific research programs, if we engage with them properly.

Third, quietism lacks a constructive procedure for dealing with metaphysical confusion when it arises. Simply saying "that's meaningless" does not help researchers who are genuinely puzzled about concepts like consciousness, intelligence, or AGI. These concepts pervade AI research; dismissing confusion about them is unhelpful.

Our productive confusion framework provides what quietism lacks: a systematic procedure for clarifying confusing concepts and, crucially, for inventing new questions that *are*

productive. Instead of dismissing confused questions, we transform them into answerable ones.

**Our position preserves quietism's skepticism while adding constructive content.** Yes, be skeptical of metaphysical claims. Yes, recognize when debates are idle. But instead of stopping there, use productive confusion: clarify the question to see if it's a language trap, idle, or genuinely testable; then invent related questions that are productive. This transforms skepticism from a stop sign into a tool for progress.

### 7.3. Synthesis

Our pragmatic position provides a middle path between realism and quietism by recommending a consequential stance in AI. Not "What is intelligence *really*?" (realism). Not "Intelligence is a meaningless word-game" (quietism). But rather: "What conceptions of intelligence enable us to think and act in productive ways?". This shift transforms metaphysics from a search for timeless truths into a practice of conceptual innovation.

## 8. Call to Action

Based on our position, we offer concrete recommendations for how AI researchers should deal with metaphysical concepts and puzzles.

### 8.1. When You Encounter Metaphysical Puzzles

**a)** Metaphysical confusion often signals important conceptual work to be done. The question "Can machines think?" seemed purely philosophical until Turing reframed it productively as an operational test. Many breakthroughs in AI began with taking seriously questions that seemed merely philosophical.

**b) Apply the productive confusion framework systematically:**

- **Clarify:** What sense are key terms being used in? How would we verify an answer?

- **Invent:** What related questions could generate new research directions, better experiments, or novel approaches?

**c) Judge proposed definitions by their consequences, not their supposed truth.** When someone proposes a definition of AGI, don't ask "Is that *really* what AGI is?" Instead ask: "What research questions does this definition enable? What benchmarks does it suggest? What predictions does it make?"

If a definition leads to concrete research programs, it has value regardless of whether it captures some Platonic

essence of AGI. If it generates no testable implications, it lacks content.

**d) Distinguish progress *on* a puzzle from progress *inspired by* it.** Our framework should not be read as licensing the claim that any empirical activity sparked by a metaphysical question counts as progress on that question. A research program is most clearly productive *with respect to the original puzzle* when its results bring conceptual clarity to the puzzle itself: by *dissolving* it (showing the question rested on a confusion), *refining* the concept (replacing a vague term with one or more sharpened explications, in the Carnapian sense), or *demonstrating* that one framing is more empirically adequate than its rivals. Research that takes a metaphysical puzzle as inspiration but yields findings orthogonal to it—new benchmarks, methods, or empirical phenomena—remains a legitimate good, but it is more honestly described as *productive divergence* than as resolution. Both modes are valuable under our framework, but conflating them invites overclaiming: announcing that a metaphysical question has been "answered" when in fact the research has simply moved past it. Honest accounting requires us to say which type of progress we have made.

### 8.2. When Reviewing or Critiquing Work

**a) Avoid rejecting papers for "definitional disagreements" alone.** If a paper uses a different definition of intelligence, consciousness, or understanding than you prefer, don't reject it solely on that basis. Definitions are tools, not truths. Instead, evaluate: does their definition lead to interesting questions, novel experiments, or useful insights?

A paper studying "intelligence as efficient learning" and a paper studying "intelligence as robust generalization" are both valuable even if they use different definitions. The criterion is whether they advance understanding, not whether they use your preferred terminology.

**b) Demand testable consequences from theoretical claims.** When papers make claims about "true intelligence" or "real understanding," ask: what observable differences would we see if this were true versus false? What experiments would confirm or disconfirm the claim? If no answer emerges, the claim lacks the empirical consequences needed to be scientifically meaningful.

This doesn't mean rejecting all theoretical work, but being aware of potential unfalsifiable assertions.

### 8.3. When Designing Research Programs

**a) Develop operational definitions that connect to measurable outcomes.** As mentioned in 5.3, rather than arguing about what AGI "really is," propose definitions that suggest specific capabilities to test, benchmarks to create, or metrics to measure (Hendrycks et al., 2025). Make your concepts

empirically tractable.

For example, instead of defining world models abstractly, specify: "We define world models for this research program as performance on benchmark X, measured by metric Y, because this operationalization enables us to test hypothesis Z." This makes the definition's purpose clear and its utility evaluable.

**b) Embrace plurality when it's productive.** As mentioned in 3.1, multiple definitions of intelligence, each emphasizing different aspects, can all be valuable if they lead to different insights. Don't insist on convergence to a single "true" definition. The field benefits from diverse perspectives.

**c) Build empirical models for thorny metaphysical puzzles.** As clarified in 5.1, don't assume that concepts like consciousness or understanding are immune to empirical investigation. Model these puzzles as frameworks that generate testable predictions. For instance, if you claim an AI system "understands" language, specify what this entails: Will it generalize in particular ways? Will it exhibit certain robustness properties? Will it fail on certain tests?

Metaphysical concepts migrate into science when the usefulness of the empirical consequences of a framework is clearly demonstrated.

### 8.4. When Communicating About AI

**a) Be explicit about the consequences of your conceptual choices.** As mentioned in 5.2, when you define terms like "scheming," explain what this definition enables you to do, what questions it helps answer, and what it might obscure. Acknowledge that other definitions serve other purposes.

This intellectual honesty prevents unproductive debates and helps readers understand your research program's assumptions.

**b) Resist unproductive anthropomorphization debates.** Whether we call an AI system's behavior "scheming," "optimizing," or "heuristic-following" should depend on which framing leads to better predictions and understanding, not on metaphysical commitments about what "truly" counts as scheming.

## 9. Conclusion

**Ultimately, we call on the AI community to adopt pragmatism as its default philosophical stance.** When metaphysical puzzles arise, approach them not as invitations to discover hidden truths, but as opportunities to invent new concepts, develop better theories, and ask more productive questions.

This shift affects how we design benchmarks, evaluate progress, communicate findings, and think about long-term research directions. By judging metaphysical concepts through their practical consequences rather than their supposed truth, we can transform unproductive philosophical debates into engines of scientific progress.

The field of AI is uniquely positioned to benefit from this pragmatic approach. Unlike physics or chemistry with centuries of accumulated theory and practice, AI still grapples with fundamental conceptual questions. We lack consensus on basic concepts like intelligence, understanding, reasoning, and learning.

Consider how this applies to current AI challenges:

- **AI Safety**: Instead of debating whether AI "really" has goals or values, ask: What behavioral patterns should we expect from systems trained with objective functions? How can we predict and prevent undesirable outcomes?

- **AI Alignment**: Instead of asking what values AI "should" have, ask: What specifications lead to reliably beneficial behavior? What training procedures produce robust alignment?

- **AI Interpretability**: Instead of asking whether we can know what AI "really" thinks, ask: What patterns in activations predict behavior? What interventions change decisions in predictable ways?

Each reformulation transforms a potentially idle philosophical question into an active research program with clear empirical consequences. Metaphysics should generate new questions, new experiments, and new ways of seeing by transforming confusion into invention, and philosophical puzzles into scientific progress. This is how metaphysics can serve AI: not by providing answers, but by provoking better questions.

## Acknowledgments

We thank our colleagues at Lossfunk and the anonymous reviewers for their substantial engagement with the paper that resulted in numerous improvements.

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
