# OpenReview forum: "Position: Metaphysical Concepts in AI Should Be Judged by Their Consequences"
_ICML.cc/2026/Position_Paper_Track — ICML 2026 Position Paper Track regular_

### Official Review · Reviewer_LUgj · 2026-03-04

**Significance:** 2
**Argument Clarity:** 4
**Rating:** 4
**Confidence:** 3

**Questions:**

In Questions 1, 2, I want to understand your thoughts and the ways your position can face limitations.

1. If a metaphysical puzzle is idle (something you acknowledge in Sec 6.2 can often be the case), is it correct to characterize your position as being content with proposing productive research questions but not necessarily taking steps towards solving the underlying puzzle?
This would be a limitation in my eyes because I believe one could end up having idle metaphysical puzzle where a concrete answer is highly consequential. For eg. "Does an AI model possess a soul?" can have major consequences in terms of society's view about them or in terms of "AI rights" but maybe any "useful" definition of "soul" leads to something idle.
How do you view the consequences of adopting your position in that case? You can answer both in the context of this specific question and more generally.

2. Are traditional metaphysical philosophies (for eg. ancient Greek/Indian philosophies) pointless in application for your position besides **inspiring** any testable models/theories (lines 297-307)? Do you think it is possible that society ends up having no other option but to rely on such philosophies and non-testable beliefs to answer crucial idle metphysical puzzles? It is not simply the idle nature of certain puzzles that can be a challenge but also the fact that we might **require** some "satisfactory" (meant in the sense of acceptable to majority of society) answer in **limited amount of time**.

3. If I were to strawman your position as simply reiterating what the whole aim of science is -- "Propose a testable theory that explains some observations and judge usefulness of the theory based upon the results of the tests", what would I be missing?

**Alternative Views Section:**

Yes

**Compliance With Llm Reviewing Policy A Conservative:**

Affirmed.

**Discussion Potential:**

3

**Paper Summary:**

The paper proposes a framework of how to apprach metaphysical questions in the context of AI. Such questions are often philosophical in nature may not lead to any testable predictions or productive outcomes. The framework, termed "Productive Confusion" pushes the idea that clarifying the nature of underlying metaphysical concept, operationalising its definitions and judging the quality of resulting consequences should be the default approach of navigating the underlying question to maximize scientific progress.

**Position:**

Yes

**Position In Title:**

Yes

**Related Work:**

3

**Strengths And Weaknesses:**

Strengths:

- The writing and paper presentation is very clear.

- I would expect this paper to inspire discussion. It is very apparent these days that due to increasing capabilities of foundation models, there are many more debates/discussions related to AI and metaphysical issues.
This paper is certainly timely in the sense of giving a reader general idea of different aspects of such discussions and more productive way of approaching them.

- The argumentation and its presentation in this paper is impressive! Given the generality of the underlying question being tackled, it was really nice to see the authors frequently supporting their arguments with multiple examples for greater precision. This theme is often encountered in philosophical texts and was well suited here.

- I do not have much background in AI philosophy literature (only ML/AI and bit of philosophy enthusiasm) so I am uncertain about this assessment, but it seemed to me the authors were comprehensive in contexualizing their work with prior literature.




Weaknesses:
- W1 (**Application**): The impact of the authors' proposal would have been much stronger if they could have applied their framework to some AI metaphysical puzzle to propose a novel research direction. As another option, an illustration to compare two contemporary approaches where no scientific consensus currently exists could also have been great.

  The closest examples in the paper are in Sec 4.2, 4.3, but the application of their framework is shown to align with concrete research questions/directions already proposed, not proposing anything novel by itself.

  It would have been amazing to see the framework in action to lead to some novel research directions/questions.

- W2: See Questions 1, 2

- W3: See Question 3

**Support:**

2

---

> ### Author Rebuttal · Authors · 2026-03-26
>
> We thank the reviewer for their feedback. We address the main concerns below.
>
> **W1 (Novel application).** We supply a worked example of our framework applied to a contemporary issue in AI: "Do LLMs have a world model?". We hope this shows both how to apply the framework and how it could lead to novel research directions.
>
> Step 1 (Clarify). We notice that the question contains two terms with multiple family resemblances, each of which could lead to a completely different interpretation. This confusion could mean researchers talking past each other and a lack of consensus on the answer.
>
> "World" could mean many things: the actual physical world we live in, the entire universe (with emphasis on physics), the abstract world an AI model encounters (such as a game or virtual environment), a particular domain (e.g. chemistry) or all possible worlds (including the world of math and logic). Similarly, "Model" could mean multiple things: symbolic representation of dynamics (such as ODEs/PDEs), subsymbolic prediction of next state, pixel-level video rollouts. Models could also emphasize different aspects: high fidelity vs low fidelity predictions, internal consistency v/s adequacy, or causality v/s correlational representations.
>
> **Because of multiple resemblances, the question "Do LLMs have a world model?" cannot be answered as stated.**
>
> Step 2 (Invent). Instead of dismissing the original question, we ask what productive research directions this puzzle could lead to. Our inspiration is all the senses in which "world" and "model" are used and then applying them to LLMs. Some examples: (a) Can LLMs predict results of novel physical experiments that are not present in their training data? (dynamics model × physical world). (b) Can LLMs answer counterfactual questions that require causal intervention rather than mere statistical conditioning? (causal model × actual world). (c) If fine-tuned on the rules of a novel toy physics never seen in pretraining, can LLMs simulate trajectories? (dynamics model × specific environment;). (d) Can LLMs detect causal inconsistencies embedded in long texts? (causal model × linguistic world).
>
> Note what the framework prevents: without the Clarify step, a researcher might test "world model" by asking an LLM to predict what happens next in a physical scenario. Success would be declared as evidence for a world model. But the clarification reveals that prediction-from-statistical-correlation and prediction-from-causal-model are different claims requiring different experiments. The framework generates not just more experiments, but better-targeted ones.
>
> > Q1 (idle but consequential puzzles)
>
> We do not dismiss metaphysical puzzles as ultimately unanswerable. Our position is that only by engaging in empirically tractable research questions can we hope to gain clarity on the metaphysical puzzles. In Section 5, our example of "What is an atom?" may serve to be helpful. When Greek philosophers asked the question, it was a metaphysical puzzle. Only later when we made empirical progress did we answer that question adequately.
>
> Hence, **for a question such as "Does an AI model possess a soul?", we do not advocate rejecting it but rather probing it by means of related empirical questions**. The Clarify step would decompose "soul" into its family resemblances: moral standing, subjective experience, continuity of identity, spiritual essence. Some of these (moral standing, subjective experience) can migrate into empirical territory; others (spiritual essence) may remain genuinely idle. Our position is that the productive components should be separated out and pursued, rather than letting the idle components paralyse progress on the tractable ones.
>
> > Q2 (traditional metaphysical philosophies)
>
> We agree that even for empirically idle questions, some answers may be psychologically more satisfactory than others and, in that way, consequentially superior to alternative views. This actually gets subsumed by our Pragmatism: instead of looking for the supposed "truth", we should look for supposed downstream consequences. When it comes to personal beliefs about empirically intractable questions, a belief that leads to relief is hence superior to one that leads to anxiety.
>
> > Q3 (Strawman: "isn't this just science?")
>
> Science presupposes operationalized concepts. **Our framework addresses the pre-scientific phase: the transition from vague metaphysical concepts to operationalized ones**. Scientific method says "propose a testable hypothesis" but does not tell you how to get from a concept like "consciousness" or "understanding" to a testable hypothesis in the first place. The Clarify step decomposes the concept into its family resemblances; the Invent step transforms those into candidate empirical questions.

---

> > ### Author Rebuttal · Reviewer_LUgj · 2026-04-02
> >
> > Thanks for the rebuttal.
> >
> > It addresses my queries reasonably satisfactorily. I am happy with the highlighted application. It is worth incorporating in the paper. Based on the rebuttal and other reviews, I would like to maintain my positive rating and favour acceptance.

---

### Official Review · Reviewer_DFCb · 2026-03-10

**Significance:** 2
**Argument Clarity:** 4
**Rating:** 4
**Confidence:** 4

**Questions:**

1. There is no related work section. Is there no work on philosophical positions in AI? How does this literature connect to the authors' work here?

2. Is the pragmatist conceptual contribution already mostly present in Kloppenberg, 1996 and others cited in the paper? It is unclear whether there might be something new emerging from the deployment of pragmatist philosophy to AI beyond its potential to organize the field.

3. Does 'consequence-evaluation' require researchers to enact the research directions? Or can they be evaluated in principle?

4. In section 3.2, the examples of questions generated by reframing incidentally coincide with the current trend of AI research, but as described before that section, it is possible to dissolve the question altogether. For instance, it might be the case that some of these questions are not worth asking. Is this a plausible scenario of deploying the authors' framework? Can you expand on this?

5. On P.4 the authors advocate for Dennet's solution to "adopt whatever interpretive framework best predicts behavior, without metaphysical commitment to what the system ’really’ has." They then immediately deploy it to the case at hand: "empirically it appears that when faced with obstacles, the model does not give up. It continues to explore alternative paths to its objectives." This description seems to this reviewer as loaded as any other. "Empirically, it appears that..." appears to misconstrue undisclosed intuitions as data-driven. The described behavior is compatible with many descriptions that contradict, for instance, that "it continues to explore alternative paths to its objectives", which is a substantial untested generalization.

6.  Throughout the paper, the authors give examples of research inventing definitions and benchmark operationalisations in support of the positive consequences of adopting the pragmatist position and the progress that ensues. However, the publication of papers, the creation of benchmarks, etc, is not itself a measure of progress. A myriad of influences affect the rate with which researchers produce new benchmarks, many of which are unrelated to scientific progress in the sense of advancing our understanding. Is it possible to give examples in support of the authors' position that tell a story about the development and advancement of knowledge rather than increase in productivity?

7. Point 3 in the authors' Call to Action proposed to judge definitions by what predictions and research directions it generates.  This call to action seems overly permissive, as it is easy to come up with definitions of AGI that generate benchmarks and other measures of field productivity without advancing any understanding of generality in cognition and computation. It appears that there are conceptual/theoretical problems that are being 'thrown out with the bath water' in the version of pragmatism advocated by the authors. Can this be clarified to strengthen the position?

8. On P.7, the authors state "Focus on behavior and capability, not on whether systems have the 'right' kind of internal states." This seems arbitrary and there is no justification for it in the pragmatist position argued in the preceding sections. Why wouldn't internal states be relevant observables useful for empirical tests?

9. Numbering of subsections and calls to action is confusing. Can you use a different numbering scheme?

**Alternative Views Section:**

Yes

**Compliance With Llm Reviewing Policy A Conservative:**

Affirmed.

**Discussion Potential:**

2

**Final Justification:**

Some of my concerns remain unresolved, which pertain to the main position. Our exchange could not find alignment on the main concern, devolving into an argument about realism versus pragmatism instead. On balance, however, I maintain my positive score based on the rest of my assessment in my original review and the rebuttals.

**Paper Summary:**

The paper argues that a pragmatist philosophical position should be the default in AI research.
This entails judgding concepts at play (e.g., AGI, consciousness) by their effect on AI research programs (e.g., new experiments, benchmarks).
This is contrasted with a Realist and a Quietist position, against which Pragmatism figures as a middle ground.
The authors elaborate the position with the notion of "productive confusion", which involves clarifying the senses in which concepts are used, and inventing variants that can make researchers think and/or act differently and productively.
Three case studies are described, involving understanding in the context of the Chinese Room, O1 'scheming', and AGI.
Finally, a 'Call to Action' section specifies concrete attitudes to adopt at each step in AI research programs.

**Position:**

Yes

**Position In Title:**

Yes

**Related Work:**

2

**Strengths And Weaknesses:**

**Strengths**
* Well written
* Arguments are easy to follow
* The topic is potentially relevant
* The position is articulated clearly

**Weaknesses**
1) The Alternative Views section discusses other positions in the philosophical literature but does not include positions coming out of the AI community.

2) There is no Related Work section or anything resembling a discussion of related position papers elsewhere

3) The position seems to be a direct application of existing positions in philosophy of science, that apply more broadly, to AI. This is not a weakness in itself, but the fact that it is unclear whether anything noteworthy comes out of its specialization to the field of AI arguably is.

4) The authors argue convincingly about Pragmatism itself, as a position on its own, but it remains unclear why one position should be preferred over the other. One could argue that Realist positions generate as many productive paths forward as any other, and in particular Pragmatism (e.g., by deriving the empirical consequences of a realist interpretation of concepts). The call to adopt Pragmatism as the default position is stated, but not argued.

5) The evidence of progress used to argue for a pragmatic perspective is somewhat superficial (e.g., conflates progress with productivity) and therefore arguably weak (production of benchmarks and empirical tests is used throughout as a stand-in for advancement of knowledge). The notion of pragmatism advocated for in the paper appears to be overly permissive, in that it allows for research programs that generate conceptual and theoretical problems without any attention payed to resolve them as long as they also generate experiments and benchmarks. This leaves the Call to Action underconstrained, and diminishes its practical value.

**Support:**

2

---

> ### Author Rebuttal · Authors · 2026-03-26
>
> We thank the reviewer for the detailed review.
>
> > On related work and alternative views within AI
>
> **We agree that the paper should contain a Related Work section and include positions arising out of the AI community itself. We will do this in our revised manuscript**. Particularly, the following papers make Pragmatist appeals towards different subdomains within AI:
>
> - AI personhood: Leibo, Joel Z., et al. "A pragmatic view of AI personhood."
> - AI safety: Nanda Neel, et al. "A Pragmatic Vision for Interpretability"
> - LLMs: Leyton-Brown, Kevin, and Yoav Shoham. "Understanding understanding: A pragmatic framework motivated by large language models."
>
> Our work differentiates from papers above by (a) justifying pragmatism by connecting it to thought within the philosophy of science; and (b) recommending an actionable framework (productive confusion) to the ML research community to apply to any metaphysical question.
>
> > On Pragmatism v/s Realism
>
> As we noted in our reply to reviewer 6bcB, **our position is compatible with realism**. In our revised manuscript we will make this clearer. We concede that Pragmatism could serve as a useful guide for enquiry even if one has realist inclinations.
>
> > On the paper's contribution beyond reiterating existing positions in philosophy of science
>
> **Our aim is to connect two established domains of enquiry (AI research and philosophy of science) which historically have not seen significant crossover**. The Pragmatist philosophy of science has a rich history going back hundreds of years. Our paper reintroduces such historical thought and makes it actionable for the AI community by means of relatable arguments and examples from within the field. More specifically, our contribution is threefold: (1) the two-step productive confusion framework provides a systematic procedure that existing pragmatist philosophy does not operationalize for working researchers; (2) AI uniquely builds the systems whose metaphysical status is debated, creating a feedback loop between conceptual and engineering work; and (3) we identify specific failure modes of how the AI community currently handles metaphysical concepts and provide concrete correctives.
>
> > On what constitutes progress (W5, Q6-Q7)
>
> We do not believe productivity is the same as progress. Hence, specific to the question of whether benchmark production constitutes progress, we remain neutral in the general case. Predicting potential progress that a research direction could lead to is very hard (often we get to know only in hindsight as many promising paths lead to dead-ends and many dead-ends suddenly bloom).
>
> **We recognise the reviewer's concern about permissiveness, however, and offer the following clarification**. Our framework does not claim that all empirical activity inspired by a metaphysical puzzle constitutes progress on that puzzle. A research direction may prove productive in unexpected ways, leading to insights orthogonal to the original question (as Searle's Chinese Room led not to an answer about understanding, but to research in knowledge representation and NLP). However, with respect to the original metaphysical puzzle, a research program is most clearly productive when its results bring clarity to the puzzle itself, whether by dissolving it, refining the concept, or demonstrating that one framing is more empirically adequate than another. We will add this nuance to the Call to Action.
>
> > On dissolving questions (Q4)
>
> Yes, **our framework can be applied recursively**. It is possible that some research directions generated in Step 2 end up getting dissolved. For example, "Are LLMs conscious?" could lead to "Do LLMs have an internal map of qualia similar to humans?". Here, the word "qualia" may render the question idle. This may then lead to a more tractable question: "Do LLM representations of colors, emotions, sound, etc. exhibit properties above and beyond what co-occurrences in their training text may suggest?". This is a feature of the framework, not a bug: it includes dissolution as a legitimate outcome.
>
> > On Dennett's solution (Q5)
>
> **We agree**. We will rephrase to describe the observed behavior neutrally before discussing alternative framings.
>
> > On internal states (Q8)
>
> **We agree that internal states are relevant observables**. What we meant was unobservable internal states (such as are sometimes posited for humans, although whether they exist is a metaphysical question in itself). We will clarify our language to reflect this.
>
> > On evaluating consequences in principle (Q3)
>
> Yes, **consequence-evaluation can be performed in principle**. One can assess whether a proposed definition opens specific experimental possibilities without enacting them. However, enacted consequences carry more evidential weight, as actual results are needed to determine whether a research program brings clarity to the original puzzle.
>
> > On numbering (Q9)
>
> We will update with a clearer numbering scheme.

---

> > ### Author Rebuttal · Reviewer_DFCb · 2026-04-02
> >
> > Thank you for the rebuttal. There are a few remaining issues:
> >
> > 1) Please indicate what your new discussion of alternative views will include, how they represent alternatives to your view, and what the discussion will be about.
> >
> > 2) Weakness 4 was not addressed. It was grouped with another reviewer's comment, but note they are different concerns. My comment was not about Pragmatism vs Realism (the latter was merely an example in my question). The concern is about lack of justification for the main Position: why should AI choose Pragmatism over other alternative positions?
> >
> > 3) Can you use the remaining reply to detail how you will fit the changes in response to the all reviews while adhering to the page limit? The changes seem substantial, given that you are adding related work and alternative views on top of other comments by myself and other reviewers.

---

### Official Review · Reviewer_6bcB · 2026-03-11

**Significance:** 4
**Argument Clarity:** 3
**Rating:** 5
**Confidence:** 4

**Questions:**

Please see above.

**Alternative Views Section:**

Yes

**Compliance With Llm Reviewing Policy A Conservative:**

Affirmed.

**Discussion Potential:**

4

**Final Justification:**

I remain in support of accepting this paper. I think it stakes out an interesting and worthwhile position. While I do not entirely agree with the position, I think it's a productive one for people to consider and discuss, which seems to me to be the central job of a position paper.

**Paper Summary:**

This paper argues for a pragmatic approach to addressing fundamenal questions in AI. The authors urge us to embrace *productive confusion* -- flexibly stating testable claims and not getting hung up on metaphysics or the variable way people use words, but also not shying away from the big questions. They contrast this position with realism (there are objective facts to discover) and quietism (the big questions are meaningless).

**Position:**

Yes

**Position In Title:**

Yes

**Related Work:**

4

**Strengths And Weaknesses:**

I am very sympathetic to this paper, since I myself want to pose the big questions in terms of testable claims. However, I may be a realist at heart, or I am at least very open-minded about the possibility of discovering objective truths.

This leads to my central criticism of the paper: it does not need to take the hard stance against realism that it takes (e.g., line 089, right column). It simply needs to reorient the realist debates towards testable claims.

This is significant in the context of the paper's running comparisons with physics. Many physicists are realists, and physicists (and philosphers of science) do debate the metaphysics of their field. For them, the rejection of Newtonian theories (line 262, right column) is a process of discovering fundamental truths -- a past theory was shown to be wrong -- not a pragmatist's game.

(Similar things could be said about the mathematical examples. There is no consensus on the metaphysical status of numbers. Oddly, Goedel apparently believed he did know their true nature, though his own results indicate that this is impossible. However, similar to the physicists, these fundamental difference seem to get in the way.)

However, my point is that nothing in the current paper actually hinges on this; the pragmaticist need not reject realism. (Quietism does need to be rejected, I think.)

I do not agree with the characterization of the Chinese Room thought experiment or its consequences. Searle took himself to be showing that, at a fundamental conceptual level, no amount of symbol manipulation (syntax) could lead to semantics (meaning). This is intended as a fundamental conceptual challenge to AI. I don't see the sense in which this sets up a language trap or false dichotomy (line 192, line 319). (I myself don't share Searle's intuition, but, in fairness to Searle, I don't think it can be brushed aside in this way.)

The 4o scheming example is also interesting. I think people want to answer the question of whether the system has an intentional stace with regard to the outcomes, since they believe that implies something about its future behavior and capabilities. Again, we can disagree about whether there is a fact of the matter here about intentions (the realist) question while still productively posing relevant empirical questions in this space. The only people we have to kick out of the room are the quietists.

**Support:**

3

---

> ### Author Rebuttal · Authors · 2026-03-26
>
> We thank the reviewer for pointing out the fact that our position does not depend on rejection of realism.
>
> We concede that Pragmatism could serve as a useful guide for enquiry even if one has realist inclinations. **The only clarification we'd like to offer here is that the strongest versions of realisms could prematurely narrow down research horizons as the emphasis on discovering "true nature" of the phenomenon under investigation could imply rejection of other fruitful avenues**. In the revised manuscript, we will soften the strong ontological claims (particularly in Section 6.3) and frame our position as methodological rather than ontological: regardless of whether mind-independent truths about intelligence exist, consequence-evaluation is the most productive way forward when empirical traction is lacking.
>
> > On the Chinese Room
>
> We take the reviewer's point that Searle's argument is a substantive claim about syntax and semantics, not merely a verbal confusion. We will revise Section 4.1 to present Searle's argument on its own terms before showing how the thought experiment's greatest value, from our pragmatic perspective, lies in the research programs it inspired (knowledge representation, NLP, sequence modeling, and so on). Our intent was not to dismiss Searle but to illustrate the productive potential of engaging with such puzzles. We acknowledge this was unclear.
>
> > On the o1 example
>
> We agree that phrases like "it continues to explore alternative paths to its objectives" carry interpretive weight beyond what the observed behavior strictly warrants. We will revise the language to describe the behavior more neutrally (e.g. "the model produced actions that circumvented the intended task pathway") before discussing how different framings of this behavior lead to different research programs.

---

> > ### Author Rebuttal · Reviewer_6bcB · 2026-04-01
> >
> > I appreciate the author response. On balance, I think I will maintain my currently positive score (an Accept recommendation).

---

### Official Review · Reviewer_ZyiM · 2026-03-13

**Significance:** 3
**Argument Clarity:** 4
**Rating:** 4
**Confidence:** 3

**Questions:**

* It would be great if the authors could answer how the ML community should evaluate whether a concept produces “productive research directions.” The argument presented in the paper is quite good; however, a clear speculation would make it even more concrete. Also, in the paper, it would be good to add how the proposed framework has already influenced empirical work or could guide specific research methodologies that the authors feel are important in the proposed perspective.

* The pre-assumption (or the underlying premise) of the proposed position may be too strong (considering our community is not philosophically rigorous). More specifically, I found some of the claims about the role of metaphysics in scientific progress are stated quite strongly and might benefit from a more nuanced discussion. For example, the relationship between truth-seeking and pragmatic utility in scientific theory choice is debated extensively in philosophy of science, and this complexity is only partially mentioned; there is a heavy possibility that a reader from our community is now aware of the entire literature in this counter, searching and mentioning those would also be helpful.

* In general, the stance taken is explaining the same things multiple times in multiple sections. Do the authors feel that stating the same things from multiple perspectives is important to make the claim/position more credible? What would be better to add is the connection to it via some existing/new empirical demonstration (would love to know the author’s view on these lines).

**Alternative Views Section:**

Yes

**Compliance With Llm Reviewing Policy A Conservative:**

Affirmed.

**Discussion Potential:**

2

**Final Justification:**

The authors partially resolved some of the concerns; a few of the initial concerns still remain, and overall, after reading the other reviews and their rebuttals, the paper is leaning towards acceptance.

**Paper Summary:**

The paper argues that debates about metaphysical concepts in artificial intelligence (such as intelligence, consciousness, understanding) are too metaphysical and should be evaluated based on their practical consequences rather than their supposed metaphysical truth (essentially pointing towards utility). The authors propose a framework called “productive confusion.” as a solution.

The framework involves two steps. First, researchers clarify the meanings and usages/utility of the underlying/stated metaphysical concepts in order to identify whether the underlying questions represent language traps, empirically tractable questions, or idle metaphysical speculation. Second, researchers should creatively reformulate such questions into new empirical research programs, conceptual frameworks, or experimental paradigms that advance ML/AI research.

**Position:**

Yes

**Position In Title:**

Yes

**Related Work:**

3

**Strengths And Weaknesses:**

Strengths:
* The paper is quite interesting to read, as it lays down a clear and well-articulated position, combining most of the philosophical takes in the philosophy of science.

* The paper beautifully combines concepts like the Chinese Room, AGI definitions, scheming in LLMs, along with the consciousness debates, to set up a good base for the position.

* The major take/benefit that the paper tries to add is a clear path for the popular ambiguous debates (about concepts such as AGI, intelligence, consciousness, and understanding), with a promise of proposing a constructive way to navigate conceptual disagreements.

Weaknesses
* One of the major issues/weaknesses of this work is that most of the stances that are taken in the paper are abstract in philosophy, rather than following an analytical philosophical approach; the paper heavily relies on arguments that are philosophical assertions. This makes the stance hard to evaluate and consider its reliability/utility for the conference community.

* One of the main claims says that metaphysical debates should be judged by consequences; the paper does not contain any counterfactual example, a thought experiment, or a position where these assumptions don’t hold, leaving an empty room for concreteness.

* Some clear directions for the ML community are also missing; it would be great if the authors would also have posted a list of fields where their argument is primarily utilized. Including some interesting takes on those lines will make it more aligned with the ML community and be more effective/impactful. Though there is a call to action section, I was expecting some more concrete steps from this section, that would be helpful for the ML community designing more experiments/benchmarks/tasks and other evaluation strategies. I believe this leaves an open end in the paper and makes it less satisfying in the end.

**Support:**

3

---

> ### Author Rebuttal · Authors · 2026-03-26
>
> Thank you for your insightful comments. We will answer your questions about how the ML community should evaluate productive research directions, and in the process, we hope some of your other concerns get addressed too.
>
> **Our position differentiates between productivity and progress**, and the paper recommends that metaphysical puzzles should be viewed as a source of inspiration for empirical research programs, some of which will ultimately lead to progress in the community. We refrain from recommending specific criteria for evaluating which directions could constitute progress because (due to the nature of generating uncertain knowledge in research) such evaluations are only possible post-facto. Throughout scientific history (and even in the ML community), what's promising and what's a dead-end could only be determined in hindsight. Deep learning and symbolic AI are some prime examples, and predicting their promise had turned out to be extraordinarily difficult.
>
> However, we do believe that productivity is a precursor for progress and hence it should be the aim of a researcher to take inspiration from metaphysical puzzles instead of dismissing them as irrelevant. For example, in our view, the success of Searle's Chinese Room puzzle isn't so much in the specific answer to the question but the entire research programs it led to (e.g. knowledge representation, NLP, sequence modeling, etc.).
>
> We recognise the reviewer's concern that our framework needs a sharper filter. **We clarify: our framework does not claim that all empirical activity inspired by a metaphysical puzzle constitutes progress on that puzzle**. A research direction may be productive in unexpected ways, leading to insights orthogonal to the original question (as Searle's Chinese Room led not to an answer about understanding, but to research in knowledge representation and NLP). However, with respect to the original metaphysical puzzle, a research program is most clearly productive when its results bring clarity to the puzzle itself, whether by dissolving it, refining the concept, or demonstrating that one framing is more empirically adequate than another. We will add this nuance to the Call to Action.
>
> As a counterfactual, consider the symbolic AI community's insistence that intelligence *is* symbol manipulation. This metaphysical commitment, treated as settled truth rather than one productive framing among several, may have contributed to the slow adoption of alternative approaches like neural networks. If the focus had been on pragmatic utility and not settling debates about the "true nature" of intelligence, the community may have explored a broader set of research programs earlier.
>
> > Some clear directions for the ML community are also missing
>
> Our "Call to Action" section contains recommendations for the ML community, but we concede that we could revise the section for additional clarity. We will do so in the revised paper. We do want to highlight that our recommendations are nuanced: we're not suggesting to ignore metaphysical puzzles, nor we're suggesting to get lost in them. Our recommendation is to engage with such puzzles earnestly with an aim of generating empirical research programs (which otherwise would be relatively hard to conceptualize).
>
> > What would be better to add is the connection to it via some existing/new empirical demonstration
>
> **We give a worked example of a current metaphysical puzzle ("Do LLMs have world models?") in our response to the reviewer LUgj.** We show how the Clarify step decomposes "world model" into its family resemblances and how the Invent step generates specific, distinct research programs from those decompositions. We hope this concrete demonstration helps clarify the nature of our recommendations.
>
> > There is a heavy possibility that a reader from our community is not aware of the entire literature
>
> We acknowledge that our paper did not cover the philosophy of science or even Pragmatism comprehensively. **The primary reason for omission is that the field is immense and we could not cover it all under page limit constraints.** However, in the revised paper, we will provide brief mentions and references for alternative positions within philosophy of science (e.g. Laudan's "Science and Values," van Fraassen's "The Scientific Image") to give entry points for interested readers.

---

> > ### Author Rebuttal · Reviewer_ZyiM · 2026-04-04
> >
> > Thank you for clarifying the distinction between productivity and progress. I guess it provides a helpful counterfactual example to consider. However, I think some of the other key concerns still remain. The notion of productivity is still not sufficiently good for practical use by researchers or reviewers, and clearer heuristics would be valuable. The paper would also benefit from including a concrete empirical case study, such as the mentioned “world model” example, to better demonstrate how the framework applies in practice. Finally, some of the philosophical claims remain somewhat strong and would benefit from additional delicate variation in meaning/tone/presentation.

---

### Decision · Program_Chairs · 2026-04-30

**Decision:**

Accept (regular)

**Comment:**

Reviewers are positive about this paper, and the authors engaged in detailed, constructive dialogue with the reviewers.